# Message source effects on rejection and costly punishment of criticism across cultures
J. Lukas Thürmer [1,2,3] ✉, Sean M. McCrea [4] & Hikari Beck[5]

Subgroups of societies evaluate information differently, leading to partisan polarization and societal rifts world-wide. Beyond mere disagreement about facts or different preferences, we identify a group-based mechanism predicting the rejection of critical messages and costly punishment of the commenter across three previously understudied and representative cultures. Our pre-registration was peer-reviewed within the Leibniz-Institute for Psychology lab-track scheme prior to data collection and, once accepted, funded. Participants ($N = 2207$) from China (collectivism, $n = 786$), Canada (individualism, $n = 666$), and Japan (honor, $n = 755$) consistently rejected criticism of their own national group that was attributed to a source from a different national group (intergroup criticism), as compared to the same criticism from within their group. These intergroup sensitivity effects were larger in China than in Canada or Japan. In Canada and Japan only, a bystander intergroup sensitivity effect emerged such that participants rejected criticism of another national group (i.e., they do not belong to) that was attributed to a source from a different national group (intergroup criticism), as compared to the same criticism from within that group. Apparently, the processes underlying this robust effect differ between cultures. We conclude that group-based message rejection contributes to societal rifts in many different cultures.

Societies are increasingly divided[1], with members of different subgroups disagreeing on basic facts such as the necessity and effectiveness of counter-pandemic measures[2], the warming of the climate, or the crowd size at a rally[3]. Research across disciplines concurs that the current levels of this polarization drive societal divides and even sparks hostilities between (sub) groups[1,3–6]. The predominantly proposed mechanisms focus on information-based differences, such as a lack of reasoning[7,8], motivated reasoning[9–11], or actual differences in personal preferences[12]. But partisan polarization may even emerge along arbitrary lines, merely by communicating sub-group members' preferences[13]. This sensitivity of opinion cascades can result in tipping points that divide an otherwise evenly distributed population into "yays" and "nays", even without the presence of initial disagreement about an issues.

In the current paper, we demonstrate such an "information-free" tipping mechanism of diverging message reception and consequent hostilities across three understudied and representative cultures: Message source effects in critical group-based communication. While group members (e.g.,

US citizens) defensively reject criticism from outgroup members (e.g., by a British citizen) as unconstructive and threatening, they are much more likely to accept the same criticism from members of their own group (e.g., by a US citizen; Intergroup Sensitivity Effect [ISE])[14,15]. One may argue that such self-report measures are weak and biased proxies of behavior[16,17], especially when it comes to costly aggression against an opponent. But to punish and refute outgroup criticism, group members even invest their own time and money[18–20], responses that are classified as hostile. Participants in these studies view identical comments across the within-group criticism and the inter-group criticism conditions, such that differences in preferences or information do not qualify as explanations for the observed effects.

Even a consistent bystander ISE emerges, such that intergroup criticism is also rejected and punished when another group is addressed (e.g., US participants responding to criticism by a British commenter about Australians)[19–22]. In other words, participants consistently invest their own resources to punish intergroup criticism of a group to which they do not belong. This finding has the alarming implication that intergroup criticism

[1]Department of Psychology, Paris-Lodron University Salzburg, Salzburg, Austria. [2]Economic Psychology Private University Seeburg Castle, Seekirchen am Wallersee, Wallersee, Austria. [3]Salzburg Center of European Union Studies, Paris-Lodron University Salzburg, Salzburg, Austria. [4]Department of Psychology, University of Wyoming, Laramie, WA, USA. [5]Department of Sociology and Human Geography, Paris-Lodron University Salzburg, Salzburg, Austria. ✉e-mail: lukas.thuermer@plus.ac.at

not only harms the parties involved but spreads throughout the communities that witness it, thereby driving societal divides. Corroborating this view, the hostile rejection of outgroup criticism has been observed in the context of key societal divides, including the rejection of the COVID vaccine[23,24], reduced engagement against climate change[25,26], and partisan animosities in the context of democratic elections[27]. Accordingly, the rejection of outside criticism represents a key candidate tipping process driving societal rifts worldwide.

The key limitation to such a conclusion is that existing studies were almost exclusively conducted in Australia, the United States, and Europe. Only about 10% of the world population lives in these areas, and their populations may differ systematically from other populations[28]. Specifically, all three commonly studied populations are highly industrialized and score high on the cultural dimension of individualism. A few studies have investigated the rejection of criticism in samples outside the United States, Australia, and Europe[29–32], but none of these studies systematically compared primarily individualistic cultural regions with regions representative of other cultural mindsets, observed costly behavioral responses, and/or considered potential cultural moderators. Humans ubiquitously cooperate[33,34], and *culture* signifies how societies organize this cooperation, including when and how to punish transgressions[34–37]. Directly related to the employed behavioral measures of costly punishment, recent research indicates that individualistic cultures prioritize monetary over psychological incentives[38]. Testing the generalizability of costly rejection of intergroup criticism thus requires carefully replicating this candidate effect beyond samples from individualistic cultures[39].

Beyond generalizability, cultural values may shed light on why the ISE occurs at all. According to the social identity perspective, the rejection of criticism aims to protect one's group[18,40]. Accordingly, the ISE should be stronger for participants who care deeply about the group criticized. Two predictions arise from this view: First, participants' reported identification with the criticized group should moderate the effect such that the ISE should be stronger for those highly identified. Second, the ISE should not emerge when another group (i.e., that participants do not belong to) is criticized. Neither of these predictions has received consistent support in past research. While some small-scale studies observed moderation effects of social identification[18,41], large-scale studies did not[19,20]. What is more, a small number of studies of the bystander ISE demonstrate even larger effects in response to criticism of groups in which participants were not a member[19–21,42]. An alternative norm perspective proposes that perceived normative deviations drive the ISE. However, even this account would not predict this larger bystander ISE. Cultures systematically vary in their attention to norms and the importance of social identification, and a systematic cultural comparison thus affords a unique opportunity to test the social mechanisms underlying the ISE.

We provide such a test, following our peer-reviewed pre-registration protocol (see "Method" for details). We selected three countries representative of the cultural dimensions of individualism (Canada), collectivism (China), and honor (Japan)[36,43]. Despite their cultural differences, these countries are quite similar in other dimensions, such as their degree of industrialization. Moreover, Japan and China both typically score high on collectivism, but Japan adheres to an honor culture to a greater degree[44,45]. The selected countries thus provide a stringent test of the consequences of three cultural mindsets, individualism, collectivism, and honor, for responses to intergroup criticism.

The three cultural mindsets have been shown to moderate basic processes such as responses to reputational threats to one's own group[46] but also responses to general norm violations[47]. In short, collectivism entails identifying similarities and fitting in, individualism entails identifying differences and sticking out, and honor entails maintaining/defending one's social standing in terms of self-worth and reputation[48]. According to a social identity perspective, collectivistic cultures could show a greater classic ISE (i.e., source effect regarding criticism targeting participant's own group) as these cultures put a greater emphasis on incurring personal costs in the service of and for defending one's group[34,49,50]. According to a norm perspective, the overall ISE (i.e., source effect independent of the criticism target) could be greater in honor cultures that endorse more violent responses to transgressions[48,51–53], especially when these transgressions are substantial (as is the case for intergroup criticism). Honor cultures seem to promote such retaliation because those engaging in it are socially rewarded[54] and insults towards one's group are taken personally[55]. Thus, response patterns across samples from these countries would provide insight into the basis of the classic and bystander ISEs. Accordingly, we registered the following hypotheses[56]:

*Pre-registered Hypothesis 1:* Consistent with the normative account, any type of intergroup criticism would result in more negative responses to the message (i.e., motive and threat) and the commenter (commenter ratings and reported anger) compared to intragroup criticism. Thus, we expected a main effect of source of the comment on these self-report and behavioral measure (i.e., lottery allocation).

Thus, we expected that the main effect of comment source generalizes across cultures, regardless of whether the participant's group or another group was the target of the criticism.

*Pre-registered Hypothesis 2:* We provided two competing hypotheses regarding culture, based on the available accounts of the ISE and cultural psychology reviewed above:

a. According to a norm account, we expect a Source × Culture interaction, such that people from Japan [honor] show a greater classic and bystander ISE (i.e., Source main effect) than participants from China [collectivistic] and Canada [individualistic].
b. According to a social identity account, we expect a Source × Target × Culture interaction, such that people from China [collectivistic] show a greater classic ISE than participants from Canada [individualistic] and Japan [honor]; this effect should not emerge for the bystander ISE.

We moreover planned the following exploratory analyses:
E1: We will compare classic and bystander ISEs between cultures
E2: We will explore a potential moderation of the ISE by perceived appropriateness of
criticizing outgroups (i.e., norms) using Johnson-Neyman analyses.
E3: We will explore a potential moderation of the ISE by national identification using Johnson–Neyman analyses. We also pre-registered equivalence tests based on a smallest effect size of interest (SESOI) between −0.200 and 0.100 (raw mean-difference) if the source effects were not significant (E4). This was not the case, and we accordingly did not perform these analyses.

In extension of these analyses, we moreover explored participants' reported cultural clusters (or mindsets) as further potential moderators. Specifically, the individualism-collectivism dimension has been supplemented by a vertical-horizontal dimension[57], yielding four clusters (vertical individualism, horizontal individualism, vertical collectivism, horizontal collectivism). While the horizontal cluster endorses principles of equality matching (e.g., equal status for everyone), the vertical cluster endorses authority ranking (e.g., merit-based status); a vertical and a horizontal orientation may be associated with collectivism (communal sharing; focus on the group) or individualism (market pricing; focus on the individual)[58,59]. Past research found relations between affective polarization and vertical individualism[5] as well as altruism and horizontal collectivism[60]. Polarization may exacerbate the group boundaries that prohibit critical debate, and typical ISE measures of costly punishment may be construed of as a form of altruistic behavior[61]. Accordingly, the endorsement of these cultural clusters could moderate the ISE, which we sought to explore in addition to our formally registered hypotheses.

To test our hypotheses, we conducted a pre-registered experiment. Participants from three different cultural regions evaluated critical comments, either about their own national group or another national group, and either voiced by a member of that same group or a different group. We also asked participants to complete measures of their cultural mindsets, intergroup criticism norm endorsement, and social identification, as exploratory moderators.

## Methods

### Transparency

A power analysis assuming a small effect of $f = 0.046$ and setting $1-\beta = 0.95$ and $\alpha = 0.05$ indicated a sample size of $N = 1559$ for detecting a mixed 3-way interaction. We sought to recruit $N = 1800$ to account for potential drop-outs, 300 men and 300 women per sampled country. This sample size would also be sufficient to establish equivalence between cultures, should no differences emerge. Our pre-registration protocol (https://psycharchives.org/en/item/74926c0c-425b-4210-816c-2b8232566fa0) was peer reviewed by external reviewers in the Leibniz foundations Lab Track scheme. Revisions were made after a first round of peer review, which were again evaluated and then accepted. The data collection of the accepted pre-registration protocol was then funded by the Leibniz Institute of Psychology (ZPID) service PsychLab. The pre-registration as well as all materials, data, and analyses are publicly available (see above and Data Availability Statement and Code Availability Statement). We report all manipulations, measures, and exclusions.

### Participants and design

We collected data between 03/17/2023 and 03/29/2023 using the professional panel provider Respondi (respondi.com) that was hired directly by the Leibniz Institute for Psychology (ZPID) service PsychLab ONLINE. The University of Salzburg internal review board approved this study (GZ 10/2020).

We used a 2 Comment Target (participants' own group vs. other group) × 2 Comment Source (criticized group vs. other group) within-participants design to manipulate comment source and comment target (i.e., all participants read four different critical comments). Within designs attain a high power and are appropriate in the current context[62], as demonstrated in previous ISE research[20]. For each participant, two comments were phrased to target their own national group and two to target one of the other national groups. One comment targeting each group was attributed to a member of the same group (ingroup source) and one comment to a member of a different national group (outgroup source; the survey software randomly matched comments to message source phrasing as well as the order of the comments). The gender of the message source was not specified. The study materials were designed in English with the help of one co-author who is an English native speaker, and then translated into Standard Mandarin Chinese by a Taiwanese native speaker as well as Japanese by another co-author who is a Japanese native speaker using a machine-aided back-translation procedure[63]. Each participant received the materials in their native language.

A total of 2462 participants completed our study (our dataset contains 2930 incomplete study entries, including those who indicated that they were not from one of the studied countries or failed attention checks and thus could not continue [see below]). Incomplete entries were fewest in Canada $n = 355$ (35%), the smallest of the three samples, and comparable in China $n = 989$ (56%) and Japan = 741 (50%).

As preregistered, we excluded 194 participants for giving the same response on at least one questionnaire page ("straight liners") and 61 for participating repeatedly (as indicated by their IP address), leaving $N = 2207$ for analysis (1245 men, 961 women, 1 other/non-binary; $M_{age} = 46.87$, $SD = 11.98$, range [17; 75]; Canada $n = 666$, China $n = 786$, Japan $n = 755$). The target quota of 300 men participants and 300 women participants was exceeded in each country (Canada men $n = 325$, women $n = 340$; China men $n = 472$, women $n = 314$; Japan men $n = 448$, women $n = 307$). Accordingly, we had a high power to test our registered hypotheses.

The order of the comments and order of the conditions were fully counterbalanced (randomized) by the survey software formR[64] and did not affect the reported results. Comment content affected some of the dependent measures, and we accordingly z-standardized all dependent measures for statistical tests. For ease of interpretation, means and standard deviations, as well as figures, are plotted with raw scale values.

### Ethics and inclusion statement

Our study design was directly adapted from a published registered report to allow for a stringent cross-cultural comparison. Native speakers who had lived in the respective cultures for large parts of their lives translated the materials to preserve the meaning while providing culturally appropriate accounts. For instance, great care was taken that the critical comments would be equally harsh but not discriminatory in the studied cultures. All translators were offered roles as co-authors according to APA and journal standards (e.g., by contributing to and confirming the final manuscript as well as potential revisions).

Our research protocol was directly adapted from studies conducted with American and Austrian populations, and the same ethical criteria were adopted for the current data collection. The study was reviewed by the Salzburg ethics review committee as the study was hosted and run by the Salzburg team. As the study was conducted online by a panel provider, contact to participants was established without a physical presence in the respective regions. Accordingly, no local IRB was required.

### Materials and procedure

Participants responded to the panel provider's study advertisements by clicking on a link to a web-based survey in form[64]. Participants provided informed consent, indicated their current location (Canada, China, Japan, other), and gender for screening purposes. We pre-registered to stratify our sample according to participant biological sex, but deviated from this choice as self-report may be better-suited to assess gender identification. This way, we were also able to retain participants with a non-binary gender identity.

Participants were then told that we needed their help in rating comments by previous participants in a study on people's thoughts about international teamwork in times of globalization, and to learn about their personal opinion. The comments were directly adapted from a previous registered report[20] to allow for reliable intercultural comparisons. Participants read critical comments describing people from the respective target country as difficult to work with. We used four such comments that were all very similar in length (62-64 words; Supplementary Information 1) and all contained negatively-valenced adjectives. Two comments targeted people from the participants' country (ingroup target condition, e.g., people from China for Chinese participants), and the other two comments targeted people from another country (outgroup target condition, e.g., people from Canada for Chinese participants). To manipulate the message source, each comment was attributed to a former participant who was described as either being from the same country as the comment target (ingroup source condition) or another country (outgroup source condition). The order of the comments, the conditions, and the two outgroup roles (target, source) was fully counterbalanced (i.e., randomized).

After reading each comment, participants indicated the nationality of the commenter and the target group of the comment, as manipulation checks, and responded to the comment. Specifically, participants rated message motive towards the target country (3 items: To what extent do you think…the comments were constructive; …the person who wrote these comments cares about [Canada, China, Japan]; …the comments were made in [Canada, China, Japan]'s best interest?), the message threat (8 items: To what extent do you think this comment is: threatening, disappointing, irritating, offensive, insulting, hypocritical, judgmental, arrogant), their evaluation of the commenter (7 items: To what extent did you find the person who wrote the comment: intelligent, trustworthy, friendly, open-minded, likable, respected, interesting), and their feelings towards the commenter (3 items: Rate how you feel about the commenter: How angry are you with the commenter? How furious are you with the commenter? How irritated are you with the commenter?). All scales were answered on seven-point scales (1: *not at all* to 7: *very much*) and adapted from previous ISE research[20,40].

After these ratings, participants had the opportunity to punish the commenter by denying lottery tickets in a drawing for a bonus in the respective currency, roughly equivalent to US$50 (C$70; JP¥7000, CN¥350). As a measure of punishment, all participants were asked how much of a bonus the commenter deserves on a 6-point slider scale (Excellent comment that deserves the full bonus of 12 tickets; Good comment that deserves a large bonus of 8 tickets; Reasonable comment that deserves a bonus of 6

tickets; Deficient comment that still deserves a small bonus of 4 tickets; Unacceptable comment that deserves 0 tickets). To make punishment costly, participants were told that they would receive 5 lottery tickets as a bonus plus half of the commenter's lottery ticket bonus for their help. Therefore, participants were able to earn between 5 and 11 lottery tickets per comment. Thus, the more lottery tickets they assigned, the more lottery tickets they received. Withholding lottery tickets can therefore be considered costly punishment[19,20]. Although all participants were correctly instructed that they were to award bonus tickets in the instructions, Chinese participants were shown cent-values (0ct to 12ct) instead of number of lottery tickets (0 tickets to 12 tickets) when making their bonus decision in Rounds 1-3; detailed analyses indicate that this programming error did not influence our results. We (a) plotted the bonus decisions per round and (b) analyzed only Round 4, where no programming error occurred (see Supplementary Information 2, for details). Both analyses were consistent with the full analysis, indicating that participants correctly assumed that they were awarding lottery tickets, as initially instructed. In line with this view, past registered reports successfully used measures with cent values and lottery tickets. Moreover, results were consistent across all other dependent measures that were displayed correctly in all rounds in all countries.

After the last comment, participants indicated their identification with their own country (4 Items: We now ask you a few questions on how important being [Canadian/Chinese/Japanese] is to you. How much do the following statements apply to you? Overall, being [Canadian/Chinese/Japanese] has very little to do with how I feel about myself. Being [Canadian/Chinese/Japanese] is an important reflection of who I am. Being [Canadian/Chinese/Japanese] is unimportant to my sense of what kind of person I am. In general, being [Canadian/Chinese/Japanese] is an important part of my self-image; adapted from Leach et al.[65]) as well as their connection to the other two countries (4 items each: I care deeply about [Canada/China/Japan]. [Canada/China/Japan] is important to me. I have a strong connection to [Canada/China/Japan]. Do you like [Canada/China/Japan]?). Participants then indicated their cultural orientation in terms of individualism/collectivism with the subscales horizontal individualism, vertical individualism, horizontal collectivism, and vertical collectivism, (4 items each, adapted from[57]), and honor (18 items, adapted from[46]) and how appropriate they found different forms of group criticism and defense (4 items: Do you condemn that people criticize other people? To what extent is it appropriate for people to: Criticize groups to which they themselves belong? Criticize groups to which they do not belong? Defend their own group from criticism by outsiders?, adapted from McCrea et al.[20]). Finally, participants reported their age, nationality, and how long they had been living in this country were thanked and fully debriefed about the deception regarding the comment sources. Participants were given the option to withdraw their consent and have their data deleted, which none of them opted to do.

## Statistical analyses

We used R[66] including the packages psych[67], readxl, expss, chron, careless[68], dplyr, effectsize[69], rstatix[70], doSNOW, foreach, lmerTest[71], jtools[72], ggplot2[73], gridExtra, gridGraphics[74], cowplot[75], broom[76], and lme4[77] to analyze our data. We computed single indexes for message motive ($\alpha = 0.89$–$0.92$), message threat ($\alpha = 0.92$–$0.93$), commenter evaluations ($\alpha = 0.96$–$0.97$), anger ($\alpha = 0.96$–$0.97$), identification with their own country ($\alpha = 0.64$; excluding items did not improve reliability; hence, all items were retained), importance of the other countries ($\alpha = 0.89$–$0.91$), the individualism-collectivism subscales (horizontal individualism $\alpha = 0.86$, vertical individualism $\alpha = 0.80$, horizontal collectivism $\alpha = 0.85$, and vertical collectivism $\alpha = 0.86$), and honor ($\alpha = 0.92$).

To test the effect of the participant nationality on the ISE, we computed generalized linear models (i.e., a mixed ANOVA procedure) using lmerTest[71] including comment source, comment target and participants' nationality, as well as the interaction term as fixed effects and each

dependent measure in turn (message motive, message threat, commenter evaluation, anger and lottery ticket allocation). The assumptions of equal variances and normality were tested, and Greenhouse–Geisser and Welch corrections were applied, if necessary. For our exploratory tests of continuous moderators, we conducted Johnson–Neyman analyses using the package interactions[78]. To account for repeatedly testing each moderator on each of the five dependent variables, we set $\alpha = 0.05/5 = 0.01$ for these exploratory analyses. Differences between cultures emerged (see below); as pre-registered, we accordingly did not perform equivalence tests.

## Reporting summary

Further information on research design is available in the Nature Portfolio Reporting Summary linked to this article.

## Results
### Confirmatory analyses

We used 2 Source (ingroup vs. outgroup; within-factor) × 2 Target (ingroup vs. outgroup; within-factor) × 3 Country (Canada vs. China vs. Japan; between-factor) mixed ANOVAs in generalized linear model procedures to test the ISE (Hypothesis 1) and explore whether it would differ between countries such that a greater overall ISE would emerge in Japan than Canada and China (Hypothesis 2a) or a greater classic ISE would emerge in China than Canada and Japan (Hypothesis 2b). In line with Hypothesis 1, we observed an overall message source effect on all dependent measures. Intergroup criticism was rated to be less constructive and more threatening than the same criticism from the ingroup (Table 1). Outgroup commenters were also rated less positively and elicited more anger than ingroup commenters (Table 1). Comment content was randomly matched to and thus identical across comment sources, and this consistent message rejection thus represents an instance of motivated reasoning. Providing evidence for behavioral consequences, outgroup commenters received fewer bonus lottery tickets than intra-group commenters (Table 1). Participants had to forego some of their own tickets to withhold these tickets, and their choice thus represents an instance of costly punishment.

We consistently observed the Source × Target × Country interactions predicted in Hypothesis 2b on all dependent measures (Table 1). We followed up on these significant interactions using post-hoc comparison tests of the source effect within each target condition, both for the overall sample and for each culture separately. The classic ISE in response to criticism of participants' own culture consistently emerged on all measures in all cultures (Table 2 and Fig. 1). The bystander ISE in response to criticism of a culture participants did not belong to consistently emerged in the Canadian and Japanese samples (Table 3 and Fig. 2). In line with past research[19,20], these bystander ISEs were descriptively larger than the observed classic ISEs. In contrast, in the Chinese sample the bystander ISE emerged only on the self-report measure of constructiveness and threat but not on commenter evaluation, anger, and the behavioral costly punishment measure bonus allocation. All these bystander effects were smaller, not larger, than the classic ISEs (Tables 2 and 3). In sum, the pattern of results in our Chinese sample is most consistent with the predictions of a social identity perspective, that is, that participants were motivated to defend their own group.

### Exploratory analyses

To compare the size of the classic ISE as well as the bystander ISE between cultures (E1), we computed respective difference scores by subtracting the ingroup source condition from the outgroup source condition and compared these values between each of the countries using independent-sample $t$-tests. We observed a significantly greater classic ISE in China than in Japan or Canada on all measures; no significant difference between Japan and Canada emerged (Table 4). The bystander ISE was significantly smaller in China than in Japan as well as Canada on all measures, except for

**Table 1 | Mixed ANOVA results for each dependent variable across the sample**

| Dependent measure | F | df | p | η² |
|---|---|---|---|---|
| **Message source** | | | | |
| Constructiveness | 526.98 | (1, 2204) | <0.001 | 0.02 |
| Message threat | 205.03 | (1, 2204) | <0.001 | 0.01 |
| Commenter evaluation | 227.64 | (1, 2204) | <0.001 | 0.01 |
| Anger | 162.00 | (1, 2204) | <0.001 | 0.01 |
| Bonus allocation | 127.88 | (1, 2204) | <0.001 | <0.01 |
| **Message target** | | | | |
| Constructiveness | 384.34 | (1, 2204) | <0.001 | 0.03 |
| Message threat | 415.43 | (1, 2204) | <0.001 | 0.04 |
| Commenter evaluation | 403.54 | (1, 2204) | <0.001 | 0.04 |
| Anger | 851.60 | (1, 2204) | <0.001 | 0.09 |
| Bonus allocation | 243.93 | (1, 2204) | <0.001 | 0.02 |
| **Participant location** | | | | |
| Constructiveness | 150.48 | (2, 2204) | <0.001 | 0.08 |
| Message threat | 66.98 | (2, 2204) | <0.001 | 0.03 |
| Commenter evaluation | 78.25 | (2, 2204) | <0.001 | 0.04 |
| Anger | 0.63 | (2, 2204) | 0.532 | <0.01 |
| Bonus allocation | 37.81 | (2, 2204) | <0.001 | 0.02 |
| **Message source × message target** | | | | |
| Constructiveness | 5.33 | (1, 2204) | 0.021 | <0.01 |
| Message threat | 0.90 | (1, 2204) | 0.344 | <0.01 |
| Commenter evaluation | 0.37 | (1, 2204) | 0.541 | <0.01 |
| Anger | 0.48 | (1, 2204) | 0.489 | <0.01 |
| Bonus allocation | 2.73 | (1, 2204) | 0.099 | <0.01 |
| **Message source × participant location** | | | | |
| Constructiveness | 2.61 | (2, 2204) | 0.074 | <0.01 |
| Message threat | 0.39 | (2, 2204) | 0.679 | <0.01 |
| Commenter evaluation | 1.14 | (2, 2204) | 0.320 | <0.01 |
| Anger | 0.43 | (2, 2204) | 0.653 | <0.01 |
| Bonus allocation | 1.95 | (2, 2204) | 0.143 | <0.01 |
| **Message target × participant location** | | | | |
| Constructiveness | 98.80 | (2, 2204) | <0.001 | 0.02 |
| Message threat | 55.59 | (2, 2204) | <0.001 | 0.01 |
| Commenter evaluation | 65.08 | (2, 2204) | <0.001 | 0.01 |
| Anger | 100.34 | (2, 2204) | <0.001 | 0.02 |
| Bonus allocation | 39.95 | (2, 2204) | <0.001 | 0.01 |
| **Message source × message target × participant location** | | | | |
| Constructiveness | 13.80 | (2, 2204) | <0.001 | <0.01 |
| Message threat | 12.57 | (2, 2204) | <0.001 | <0.01 |
| Commenter evaluation | 38.98 | (2, 2204) | <0.001 | <0.01 |
| Anger | 27.02 | (2, 2204) | <0.001 | <0.01 |
| Bonus allocation | 24.11 | (2, 2204) | <0.001 | <0.01 |

**Table 2 | Classic ISE effect sizes on each dependent variable across the full sample as well as subsamples**

| Dependent measure | t | df | p | d | 95% CI |
|---|---|---|---|---|---|
| **Full Sample** | | | | | |
| Constructiveness | 14.90 | 2206 | <0.001 | 0.32 | [0.28, 0.36] |
| Message threat | −9.03 | 2206 | <0.001 | −0.19 | [−0.23, −0.15] |
| Commenter evaluation | 11.60 | 2206 | <0.001 | 0.25 | [0.21, 0.28] |
| Anger | −9.63 | 2206 | <0.001 | −0.21 | [−0.25, −0.17] |
| Bonus allocation | 9.20 | 2206 | <0.001 | 0.20 | [0.15, 0.24] |
| **Canada** | | | | | |
| Constructiveness | 8.40 | 666 | <0.001 | 0.33 | [0.26, 0.39] |
| Message threat | −3.95 | 666 | <0.001 | −0.15 | [−0.23, −0.08] |
| Commenter evaluation | 4.97 | 666 | <0.001 | 0.19 | [0.12, 0.26] |
| Anger | −2.35 | 666 | 0.019 | −0.09 | [−0.17, −0.02] |
| Bonus allocation | 4.37 | 666 | <0.001 | 0.17 | [0.09, 0.24] |
| **China** | | | | | |
| Constructiveness | 10.10 | 786 | <0.001 | 0.36 | [0.30, 0.43] |
| Message threat | −6.84 | 786 | <0.001 | −0.24 | [−0.32, −0.17] |
| Commenter evaluation | 9.56 | 786 | <0.001 | 0.34 | [0.27, 0.40] |
| Anger | −7.64 | 786 | <0.001 | −0.27 | [−0.34, −0.20] |
| Bonus allocation | 7.74 | 786 | <0.001 | 0.28 | [0.21, 0.35] |
| **Japan** | | | | | |
| Constructiveness | 7.54 | 755 | <0.001 | 0.27 | [0.20, 0.35] |
| Message threat | −4.46 | 755 | <0.001 | −0.16 | [−0.25, −0.09] |
| Commenter evaluation | 4.63 | 755 | <0.001 | 0.17 | [0.11, 0.23] |
| Anger | −6.09 | 755 | <0.001 | −0.22 | [−0.29, −0.15] |
| Bonus allocation | 2.95 | 755 | 0.003 | 0.11 | [0.03, 0.19] |

The classic ISE is the message source effect for messages targeting participants' own group.

this end, we split the sample by Target to test the classic and bystander ISEs separately and conducted exploratory Johnson-Neyman analyses. For the classic ISE, no interactions with the message source emerged (Table SI1) and no meaningful transition points (i.e., within the observed range of the moderator) were identified (Fig. SI2a). The source effect was thus significant across the observed values of the moderator. For the bystander ISE, significant Norm × Source interactions emerged for anger and bonus allocation (Table SI2), and transition points indicated that the source effect was only significant among those with moderate or low norm endorsement (5.61 or below for anger and 5.29 or below for bonus allocation; Supplementary Information 2 and Fig. SI2b). Parallel analyses for the other dependent measures showed no significant Norm × Source interactions or meaningful transition points (i.e., within the possible scale values).

Using parallel analyses, we explored national identification as a potential moderator (E3). This analysis yielded no significant interactions (Supplementary Information 2, Tables SI3 and SI4). Johnson-Neyman Intervals indicated some transition points within the range of observed moderator values. The classic ISE on threat ratings, anger ratings, and bonus allocations was only significant for participants reporting moderate-to high identification (above 2.41, 1.69, and 1.76, respectively; Fig. SI3a). The bystander ISE on bonus allocations was not significant for those extremely highly identified (above 6.81 on a 7-point scale; Fig. SI3b). Due to the non-significant interactions, we interpret these exploratory results cautiously. Parallel analyses for the other dependent measures showed no meaningful transition points (i.e., within the possible scale values).

Given the observed cultural differences in our pre-registered analyses, we additionally sought to explore self-reported cultural moderators (honor, vertical collectivism, horizontal collectivism,

constructiveness (Table 5). Again, no significant differences emerged between Japan and Canada. In sum, these exploratory analyses corroborate our interpretation that Chinese participants responded more strongly to criticism of their own group (classic ISE) but less strongly to criticism of another group (bystander ISE), as compared to participants from Japan or Canada.

As pre-registered, we explored normative evaluations of the appropriateness of outgroup criticism as a potential moderator of the ISE (E2). To

vertical individualism, horizontal individualism). Following the procedures reported by McCrea et al.[20], we first regressed all cultural mindset measures on the classic ISE difference score and then on the

bystander ISE difference score. Vertical individualism, the support for independently striving for one's place in the hierarchy, emerged as a consistent predictor in both analyses, except for one non-

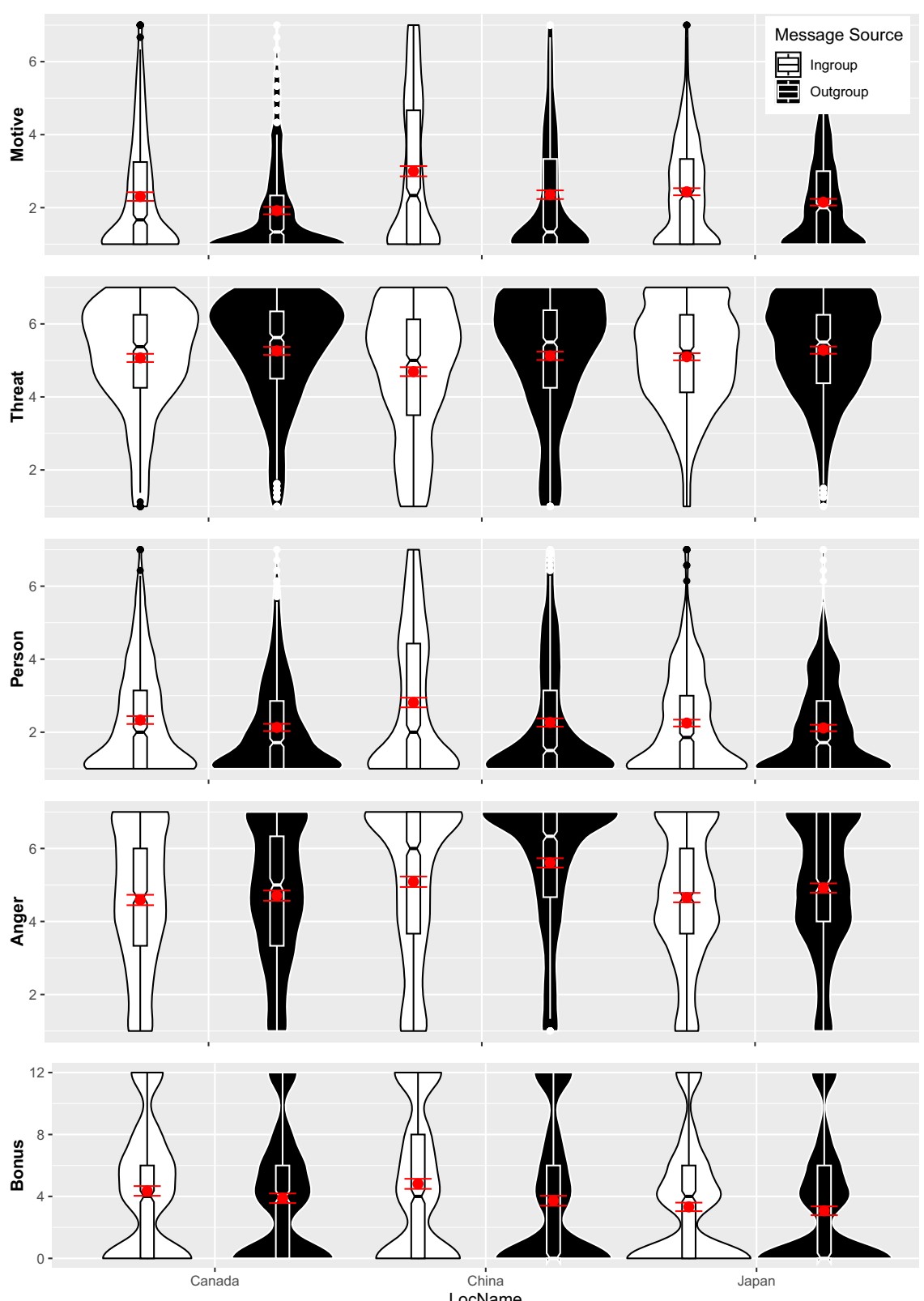

**Fig. 1 | Boxplots with Violin-Plots: responses to criticism of participants' own national group (classic ISE) by message source (ingroup vs. outgroup) and participant location.** China (collectivism, *n* = 786), Canada (individualism, *n* = 666), and Japan (honor, *n* = 755). Red dots with lines indicate means with 95% CIs. Bonus: Lottery tickets awarded to commenter.

**Table 3 | Bystander ISE effect sizes on each dependent variable across the full sample as well as subsamples**

| Dependent measure | *t* | *df* | *p* | *d* | 95% CI |
|---|---|---|---|---|---|
| **Full sample** | | | | | |
| Constructiveness | 18.20 | 2206 | <0.001 | 0.39 | [0.34, 0.43] |
| Message threat | −11.40 | 2206 | <0.001 | −0.24 | [−0.29, −0.20] |
| Commenter evaluation | 10.00 | 2206 | <0.001 | 0.21 | [0.17, 0.26] |
| Anger | −8.39 | 2206 | <0.001 | −0.18 | [−0.22, −0.13] |
| Bonus allocation | 6.89 | 2206 | <0.001 | 0.15 | [0.11, 0.19] |
| **Canada** | | | | | |
| Constructiveness | 11.30 | 666 | <0.001 | 0.43 | [0.37, 0.51] |
| Message threat | −7.95 | 666 | <0.001 | −0.31 | [−0.38, −0.24] |
| Commenter evaluation | 8.22 | 666 | <0.001 | 0.32 | [0.24, 0.40] |
| Anger | −7.18 | 666 | <0.001 | −0.28 | [−0.36, −0.20] |
| Bonus allocation | 6.64 | 666 | <0.001 | 0.26 | [0.18, 0.32] |
| **China** | | | | | |
| Constructiveness | 7.89 | 786 | <0.001 | 0.28 | [0.21, 0.36] |
| Message threat | −3.89 | 786 | 0.001 | −0.14 | [−0.21, −0.06] |
| Commenter evaluation | 1.64 | 786 | 0.102 | 0.06 | [−0.01, 0.13] |
| Anger | −0.75 | 786 | 0.454 | −0.03 | [−0.10, −0.21] |
| Bonus allocation | 0.60 | 786 | 0.552 | 0.02 | [−0.05, 0.10] |
| **Japan** | | | | | |
| Constructiveness | 13.90 | 755 | <0.001 | 0.51 | [0.44, 0.58] |
| Message threat | −8.57 | 755 | <0.001 | −0.31 | [−0.39, −0.24] |
| Commenter evaluation | 9.37 | 755 | <0.001 | 0.34 | [0.27, 0.41] |
| Anger | −7.78 | 755 | <0.001 | −0.28 | [−0.36, −0.21] |
| Bonus allocation | 5.50 | 755 | <0.001 | 0.20 | [0.12, 0.27] |

The bystander ISE is the message source effect for messages targeting a group participants do not belong to

significant effect on the constructiveness measure when another group was criticized (Tables SI5 and SI6). We then probed vertical individualism using analyses parallel to the pre-registered analyses on E2 and E3 (Tables SI7 and SI8). When participants' group was criticized, the message source effect was not significant among those low in vertical individualism (transition points: constructiveness 1.60; threat 2.98; commenter evaluation 2.86; anger 2.80; bonus allocation 2.99; Fig. 3); when another group was criticized, the message source effect was not significant among those very high in vertical individualism (transition points: constructiveness: none, threat 6.79; commenter evaluation 6.52; anger 6.15; bonus allocation 5.40; Fig. 4). All other measures related to culture did not predict the source effects on any of the dependent measures (Tables SI5 and SI6).

As an additional exploratory analysis, we included participant gender as a factor. This yielded no consistent influence on the observed Target and Source effects (see Supplementary Information 3 and Table SI9). Only one small but significant Target × Participant Gender interaction emerged on message threat ratings. In sum, we observed little evidence for gender differences in the ISE.

## Discussion

In a pre-registered experiment using peer-reviewed procedures, we systematically test for cultural variation in the costly rejection of intergroup criticism. We sampled three cultures representative of the key cultural dimensions of individualism, collectivism, and honor to conduct a highly controlled behavioral experiment. Participants in all three cultures rejected criticism of their own national group that was attributed to a source from another national group, as compared to the same comments from within their group (i.e., a classic ISE emerged). Canadian and Japanese participants

even rejected intergroup criticism of another national group they did not belong to, but this bystander ISE was not observed among Chinese participants. These results extended to our measure of costly punishment (withholding lottery tickets by giving up some of their own tickets), a common behavioral measure of hostility.

Exploratory analyses indicated that the rejection of intergroup criticism was tied to reported levels of vertical individualism, a cultural orientation supportive of individual attainments and distinction. Specifically, vertical individualism seemed to highlight the need to defend one's group and reduce the need to defend another group. Interestingly, no parallel effects emerged for vertical collectivism or horizontal individualism. This indicates that the specific combination of striving for attainment/hierarchy and doing so individually is related to the observed effects.

### Rejection of intergroup criticism causes hostile responses

Participants were willing to pay to punish outgroup commenters, a behavior that can be classified as hostile. The rejection of outgroups, as commonly observed in politically polarized environments, has been linked to violence[79]. However, a common assumption in this literature is that identification with one's own group causes these effects. Our exploratory analyses provide only inconsistent evidence that reported identification influences the rejection of outgroup criticism (see Supplementary Information 2, and Fig. SI2). What is more, participants in Canada and Japan also rejected criticism targeting other groups they did not belong to. Apparently, the mere categorization into "us" and "them" was sufficient to elicit the hostile rejection of criticism, even for those only witnessing critical exchanges. Apparently, intergroup criticism drives hostile responses not only among those highly identified and directly involved in critical exchanges.

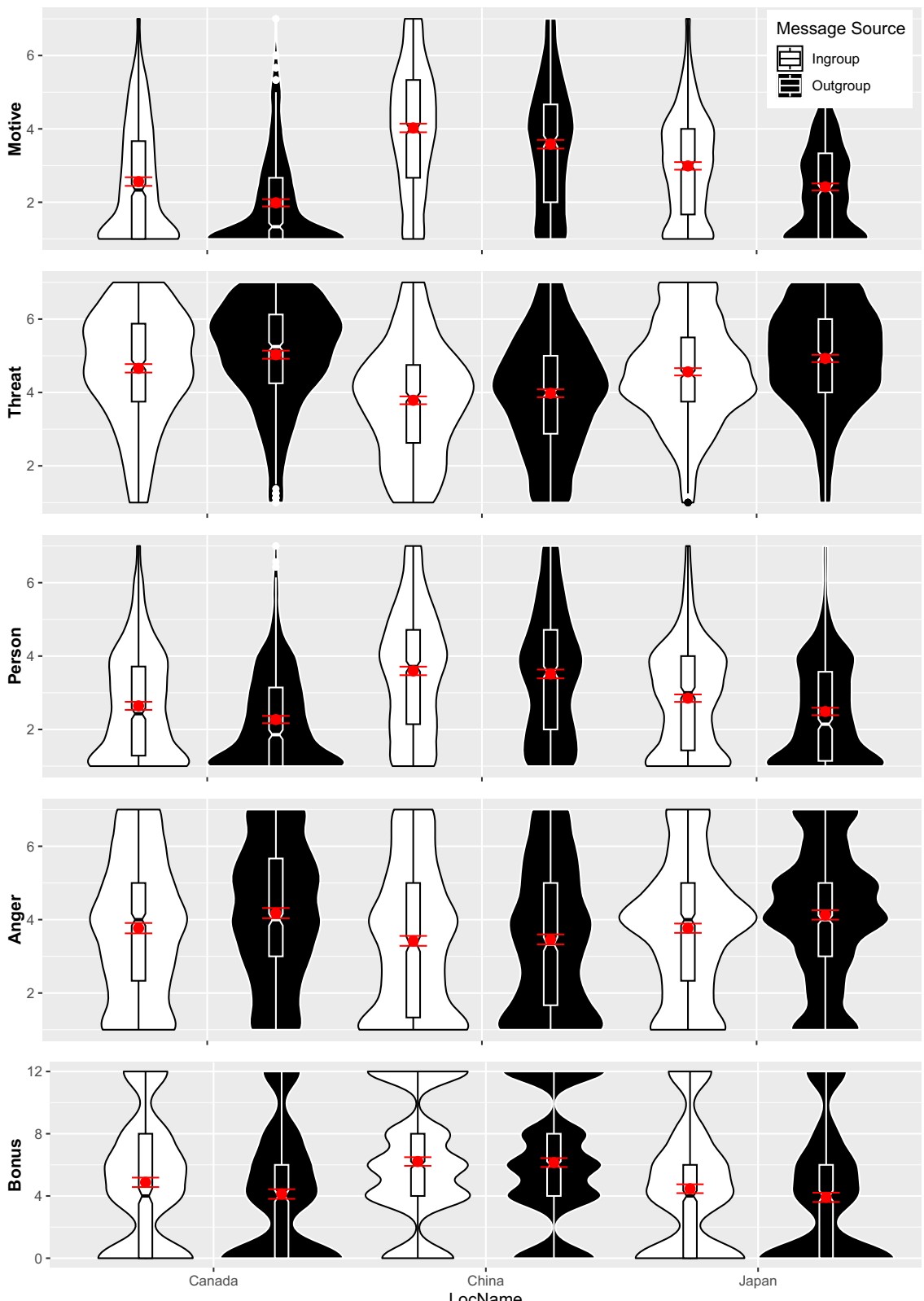

**Fig. 2 | Boxplots with Violin-Plots: responses to criticism of a national group participants did not belong to (bystander ISE) by message source (ingroup vs. outgroup) and participant location.** China (collectivism, $n = 786$), Canada (individualism, $n = 666$), and Japan (honor, $n = 755$). Red dots with lines indicate means with 95% CIs. Bonus: Lottery tickets awarded to commenter.

**Table 4 | Differences in classic ISE effect sizes on each dependent variable between cultures**

| Dependent Measure | n1, n2 | p | d | 95% CI |
|---|---|---|---|---|
| Canada-China | | | | |
| Constructiveness | (666, 786) | 0.001 | −0.17 | [−0.28, −0.07] |
| Message Threat | (666, 786) | 0.001 | 0.16 | [0.05, 0.25] |
| Commenter Evaluation | (666, 786) | <0.001 | −0.26 | [−0.36, −0.15] |
| Anger | (666, 786) | <0.001 | 0.25 | [0.15, 0.35] |
| Bonus Allocation | (666, 786) | <0.001 | 0.17 | [0.07, 0.27] |
| Canada-Japan | | | | |
| Constructiveness | (666, 755) | 0.557 | 0.09 | [−0.02, 0.19] |
| Message Threat | (666, 755) | 0.884 | −0.01 | [−0.12, 0.09] |
| Commenter Evaluation | (666, 755) | 0.294 | 0.07 | [−0.03, 0.18] |
| Anger | (666, 755) | 0.076 | 0.12 | [0.01, 0.22] |
| Bonus Allocation | (666, 755) | 0.068 | −0.12 | [−0.22, −0.02] |
| China-Japan | | | | |
| Constructiveness | (786, 755) | <0.001 | 0.24 | [0.15, 0.34] |
| Message Threat | (786, 755) | 0.002 | −0.17 | [−0.27, −0.07] |
| Commenter Evaluation | (786, 755) | <0.001 | 0.32 | [0.23, 0.42] |
| Anger | (786, 755) | 0.001 | −0.16 | [−0.26, −0.06] |
| Bonus Allocation | (786, 755) | <0.001 | −0.27 | [−0.38, −0.17] |

The classic ISE is the message source effect for messages targeting participants' own group.

**Table 5 | Differences in bystander ISE effect sizes on each dependent variable between cultures**

| Dependent Measure | n1, n2 | p | d | 95% CI |
|---|---|---|---|---|
| Canada-China | | | | |
| Constructiveness | (666, 786) | 0.060 | 0.09 | [−0.01, 0.20] |
| Message Threat | (666, 786) | 0.006 | −0.14 | [−0.24, −0.04] |
| Commenter Evaluation | (666, 786) | <0.001 | 0.22 | [0.13, 0.33] |
| Anger | (666, 786) | <0.001 | −0.24 | [−0.34, −0.13] |
| Bonus Allocation | (666, 786) | <0.001 | −0.23 | [−0.33, −0.12] |
| Canada-Japan | | | | |
| Constructiveness | (666, 755) | 0.934 | <0.01 | [−0.10, 0.11] |
| Message Threat | (666, 755) | 0.899 | −0.01 | [−0.11, 0.09] |
| Commenter Evaluation | (666, 755) | 0.868 | 0.01 | [−0.10, 0.12] |
| Anger | (666, 755) | 0.529 | −0.04 | [−0.14, 0.07] |
| Bonus Allocation | (666, 755) | 0.117 | −0.09 | [−0.19, 0.01] |
| China-Japan | | | | |
| Constructiveness | (786, 755) | 0.063 | −0.09 | [−0.20, 0.01] |
| Message Threat | (786, 755) | 0.007 | 0.14 | [0.04, 0.24] |
| Commenter Evaluation | (786, 755) | <0.001 | −0.22 | [−0.32, −0.13] |
| Anger | (786, 755) | <0.001 | 0.22 | [0.12, 0.32] |
| Bonus Allocation | (786, 755) | 0.002 | 0.16 | [0.05, 0.26] |

The bystander ISE is the message source effect for messages targeting a group participants do not belong to.

## Ubiquitous processes and cultural differences

A host of research has focused on cultural differences, for instance, on what constitutes morally appropriate behavior[80], personality differences[81], or values[82]. Other work, traditionally referred to as "basic research", has assumed that their observed causal processes are culturally invariant and accordingly recruited easily accessible samples[28,83]. And even within populations, preferences for certain types of information may widely differ[84,85]. Here, we put a candidate basic process to a critical intercultural test. The observed commonalities, namely the consistent rejection of intergroup criticism targeted at one's group, speak to a considerable overlap between the three studied cultures. The rejection of intergroup criticism thus occurs across a wide range of cultures and is a prime candidate for a psychological universal[86,87].

At the same time, we observed differences in responses to criticism targeted at another group. This finding suggests that the divisive power of intergroup criticism may be confined to the involved parties in collectivistic cultures, but it may spread to uninvolved bystanders in individualistic and honor cultures. Exploratory analyses indicated that vertical individualism, an orientation that emphasizes standing out by demonstrating one's own achievement, was related to a greater rejection of intergroup criticism of their own group but somewhat weaker rejection of intergroup criticism of other groups. One interpretation of this effect is that individuals high on this dimension are most suspicious of the motives of an intergroup commenter as someone who is attempting to push ahead of themselves and thus needs to be stopped. Malicious intergroup criticism of another group may, however, serve the goal to enhance their own or their group's status. Corroborating this view, observed affective polarization is typically greatest in countries scoring high on vertical individualism[5]. Also in line with this view, the same comments used in the current study yielded substantially larger ISEs in a registered report with US-American participants[20], a population typically scoring high on vertical individualism. Interestingly, although horizontal collectivism has been associated with altruism[60] and the costly punishment assessed in this study is sometimes referred to as altruistic punishment[61], horizontal collectivism did not emerge as a consistent moderator.

## Limitations

The current research sampled three cultures. Although these cultures are representative of the key dimensions of individualism, collectivism, and honor, other cultural dimensions may play a role. The three studied countries have also been found to differ from each other in terms of their values[83], especially in how much attention is given to the needs of others (i.e., social mindfulness)[88] and how important long-term strivings are (i.e., flexibility vs. monumentalism)[89]. The specific levels of social mindfulness (Japan #1, China #18, Canada #24 out of 31 countries studied) and flexibility (Japan #1, China #5, Canada #20 out of 54 countries studied) in the studied countries indicates that this dimension is unlikely to explain the observed effects: Participants from Canada and Japan showed similar responses to criticism, although these populations differ greatly in mindfulness. Moreover, Japan adheres to an honor (as well as face) logic where positive self-views depend on one's evaluation by others in established hierarchies that are largely cooperative[45]. Finally, Japanese participants have been observed to be relatively open to criticism, even self-criticize, apparently as a means to self-improvement[90]. In contrast, Middle-Eastern and North-African cultures adhere to an honor logic where positive self-views depend on one's evaluation by others in competitive interactions. Studying the ISE in such honor contexts is a fruitful avenue for future research.

We presented harsh, critical comments that were identical across national populations to afford a high experimental control. Despite this highly conservative test, we observed consistent effects. One may argue that effects may even be more pronounced if comments were tailored to common stereotypes of that specific national group (e.g., portraying Americans as uncultured or Europeans as old-fashioned[41]). In fact, due to their generic character, the comments used in the present study may be perceived as serving the pragmatic function of derogating the targeted group (i.e., as insults[91,92]). In line with this view, the observed ISEs in this study had descriptively smaller effect sizes than those observed in past registered reports[19,20]. It should be noted, however, that consistent message source

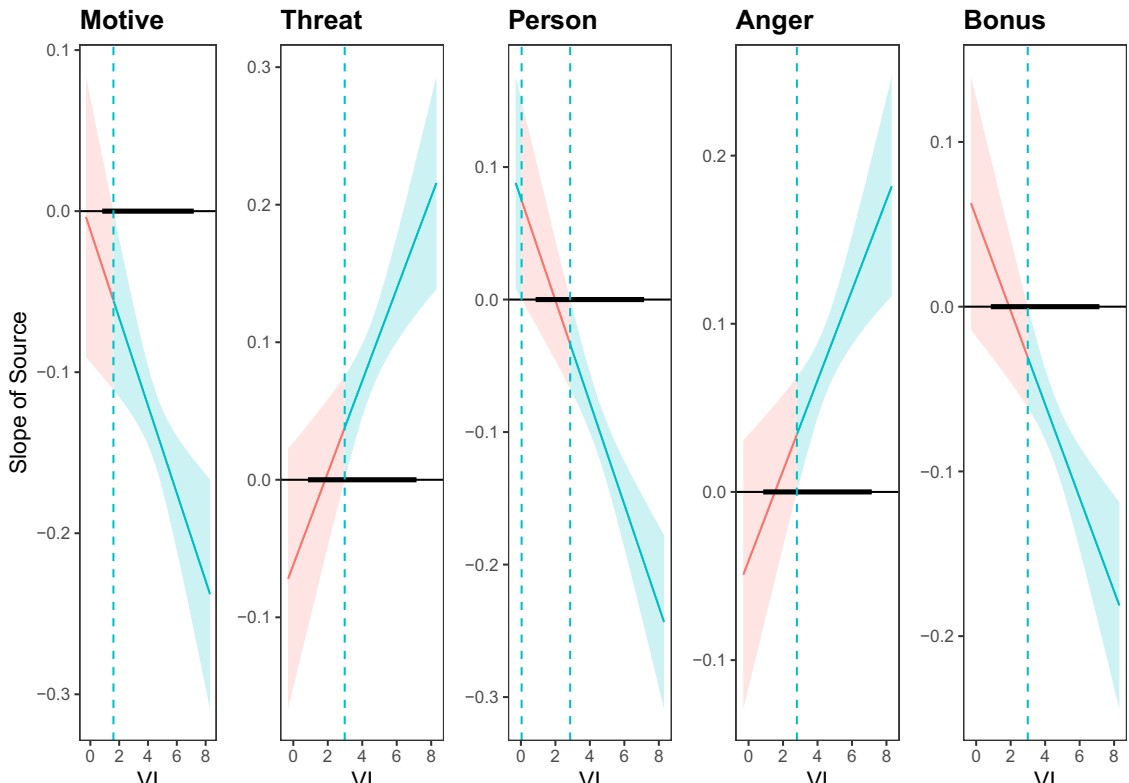

**Fig. 3 | Johnson-Neyman-Plots for the moderation analysis of the message source effect (ingroup vs. outgroup) responding to criticism of participants' own national group (classic ISE) conditional on reported vertical individualism [VI].** $N = 2207$. The message source effect (difference between within-group source and between-group source) is plotted on the $y$-axis and the moderator (vertical individualism [VI]) is plotted on the $x$-axis. The message source effect is significant at $\alpha = 0.01$ in green-shaded areas and non-significant in red-shaded areas. Bold horizontal line indicates observed values of vertical individualism. Bonus: Number of bonus lottery tickets assigned.

effects emerged nevertheless, pointing to the potential of interpreting the comments to be somewhat constructive. Exploring how differences in comment content affect responses to criticism clearly is a fruitful direction for future research[93].

One may argue that people naturally refrain from harsh criticism, as employed in the current experiment, and instead engage in more neutral conversation. Corroborating this view, other research observed that discussing divisive topics was ineffective to reduce affective polarization, but mundane day-to-day interaction had an effect[94]. Unfortunately, current societal interactions, such as on social media, prioritize inflammatory messages[95] and incentivize polarization and outrage[96,97]. It is therefore quite likely that people do encounter critical comments.

Lastly, the exploratory moderators identification and cultural preferences were assessed after the dependent measures. It is thus possible that the experimental tasks influenced these measures. However, in our within-participants design, all participants viewed all comments from each comment source. Accordingly, it is unlikely that the experimental manipulations systematically influenced the measured moderators. To empirically rule out this possibility, future studies should employ 2-stage designs, measuring the moderators at T1 and assessing responses to experimental manipulations at T2.

**Applying the rejection of criticism to societal rifts**
The hostile rejection of intergroup criticism may be conceived of as one mechanism contributing to societal rifts worldwide. In morally-laden discourse, a conflict-supporting mindset[98] has been observed to contribute to intergroup hostilities. Here, we observed that all it took for hostile responses to emerge was one critical comment across group lines. For instance, we consistently observed that participants attributed unconstructive motives to outgroup commenters. Such motive attributions may be construed of as an aspect of trust, and lacking trust is a key determinant of polarization[99]. Perhaps, conflict-supporting mindsets are engrained in our social operating system and can lead to societal tipping points much more easily than previously assumed[13].

At the same time, our research indicates that individual characteristics and preferences are imperfect predictors of message reception and behavioral responses. In line with this view, political campaigns often prove largely ineffective[100] and well-proven interventions fail in social contexts[101]. And even when treatments influence reported attitudes and intentions they may fail to influence behavior[102]. Instead, the current findings indicate that group processes are a key driver of message reception and consequent responses. In essence, the immediate rejection of outgroup criticism impedes human progress.

The current work contributes to understanding and predicting these mechanics so we can change what lies ahead. Specifically, we observed that participants from regions associated with collectivism "stayed in their lane," that is showed a message source effect in response to criticism targeting their own group (i.e., a large classic ISE) but not other groups (i.e., no bystander ISE emerged). Moreover, individuals reporting a high endorsement of vertical individualism, a cultural orientation towards hierarchical self-reliance, showed stronger rejection of outgroup criticism. This would suggest that promoting more communal and horizontal orientations should reduce the observed ISE. This speculation is in line with recent research showing that communal behaviors (apologizing for violating conversational norms and waiting to be invited to speak) reduced the ISE[24,103]. However, even under these conditions, significant source effects remained. Apparently, the rejection of intergroup criticism is a ubiquitous phenomenon that is hard to overcome.

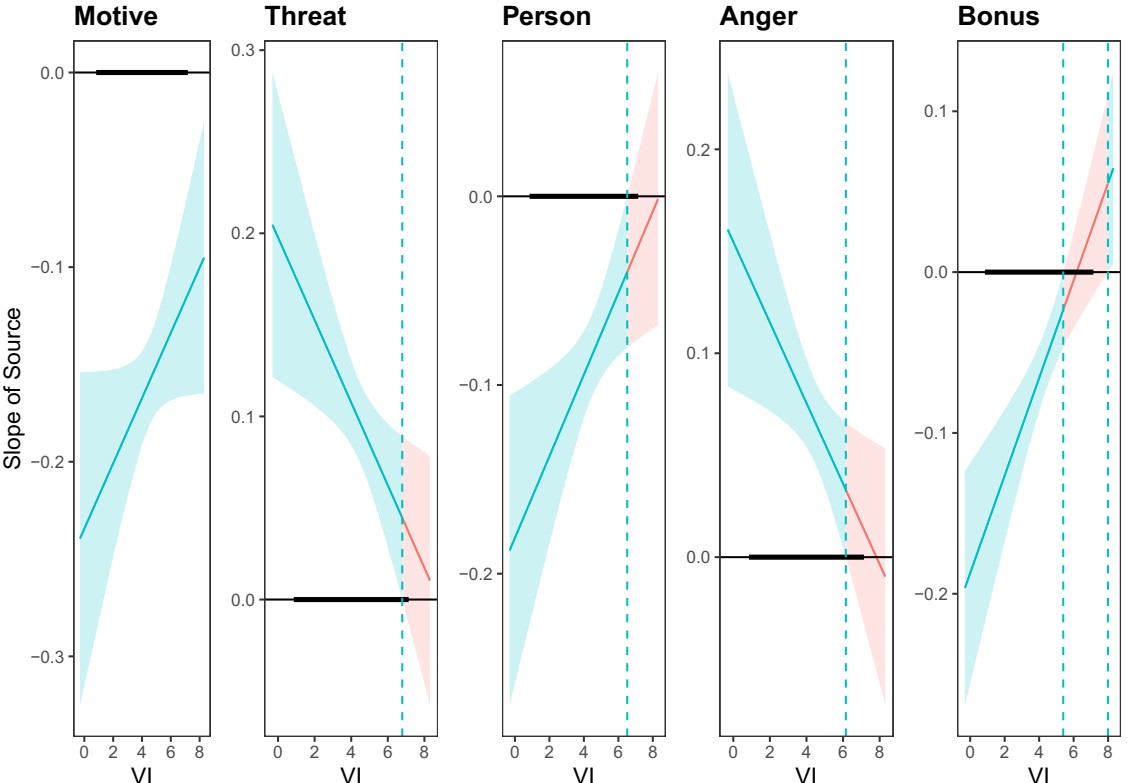

**Fig. 4 | Johnson-Neyman-Plots for the moderation analysis of the message source effect (ingroup vs. outgroup) responding to criticism of a national group participants did not belong to (bystander ISE) conditional on reported vertical individualism [VI].** $N = 2207$. The message source effect (difference between within-group source and between-group source) is plotted on the $y$-axis and the moderator (vertical individualism [VI]) is plotted on the $x$-axis. The message source effect is significant at $\alpha = 0.01$ in green-shaded areas and non-significant in red-shaded areas. Bold horizontal line indicates observed values of vertical individualism. Bonus: Number of bonus lottery tickets assigned.

## Data availability

Anonymized raw data are available at https://osf.io/djkz8/.

## Code availability

Code for analyses is available at https://osf.io/djkz8/.

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

## Acknowledgements

Data collection of the present study was funded by PsychLab, a service of the Leibniz Institute for Psychology (ZPID). This research was funded in part by the Austrian Science Fund (FWF) [https://doi.org/10.55776/P37261]. ZPID organized the peer-review of our pre-registration protocol and funded the data collection. Apart from that, the funders had no role in study design, data collection and analysis, decision to publish or preparation of the manuscript. We thank Manuela Wagner and Jakob Reichert for their assistance in conducting this research, Julia Prohaska for her assistance in preparing the manuscript, Kyoko Shinozaki for her guidance regarding cultural matters, and Wei Wei Chang-Albert for their assistance translating the materials.

## Author contributions

Conceptualization, J.L.T. and S.M.M.; translation, H.B.; methodology, J.L.T.; formal analysis, J.L.T.; resources, J.L.T.; data curation, J.L.T.; writing—original draft preparation, J.L.T.; writing—review and editing, S.M.M. and H.B.; visualization, J.L.T.; supervision, J.L.T.; project administration, J.L.T.; funding acquisition, J.L.T. All authors have read and agreed to the published version of the manuscript.

## Competing interests

The authors declare no competing interests.
