## [Transparent Peer Review file · Communications Psychology]

Message Source Effects on Rejection and Costly Punishment of Criticism Across Cultures

Corresponding Author: Dr J. Lukas Thürmer

Version 0:

Decision Letter:

Dear Dr Thürmer,

Thank you for your patience during the peer-review process. I am sorry for the significant delay.

Your manuscript titled "Message Source Effects on Rejection and Costly Punishment of Criticism Across Cultures" has now been seen by 2 experts, and I include their comments at the end of this message. They find your work of interest but raised some important points. We are interested in the possibility of publishing your study in Communications Psychology, but would like to consider your responses to these concerns and assess a revised manuscript before we make a final decision on publication.

We therefore invite you to revise and resubmit your manuscript, along with a point-by-point response to the reviewers. Please highlight all changes in the manuscript text file.

We ask that you address all referee comments through suitable revisions and in particular engage with concerns about the design and those regarding terminology (including the titular "bystander effect"). Please also include an Ethics and Inclusion statement (<https://www.nature.com/commspsychol/editorial-policies/authorship#authorship-inclusion-and-ethics-in-global-research>), addressing especially points 1 and 5.

I am attaching an Editorial Requests Table that details critical reporting requirements for the revised manuscript. Please attend to each item and ensure your manuscript is fully compliant. We are requesting that your manuscript aligns with these requirements as this facilitates the evaluation of your manuscript, reducing delays in re-review and potential future acceptance. If your revised manuscript is not aligned with these requests on major issues, such as those concerning statistics, it may be returned to you for further revisions without re-review. Additional information can be found in our style and formatting guide <https://www.nature.com/documents/commspsychol-style-formatting-guide-accept.pdf>>Communications Psychology formatting guide.

Please use the following link to submit your

- revised manuscript,
- point-by-point response to the referees' comments,
- cover letter (as a separate document),
- the Editorial Policy Checklist (see below),
- the Reporting Summary (see below), and
- the completed Editorial Request Table (attached):

Link Redacted

Best regards,

Marike

Marike Schiffer, PhD
Chief Editor
Communications Psychology

REVIEWER EXPERTISE:

Reviewer #1 behavioural economics
Reviewer #2 cross-cultural psychology

REVIEWER REPORTS:

Reviewer #1 (Remarks to the Author):

The paper explores how people across different cultures react to criticism of their national group when the criticism comes from within their group (intragroup) versus from outside their group (intergroup). The study was conducted with participants from China, Canada, and Japan, representing collectivistic, individualistic, and honor cultures, respectively. The researchers found that people generally reject criticism of their group more strongly when it comes from an outsider. This rejection also extends to criticism of other national groups, particularly in Canada and Japan.

The study is pre-registered, methodologically sound, and the analysis is well-developed. As someone coming from a different field, my take is that this study makes a significant contribution by comparing reactions across three different cultures in a high-powered study, exploring both direct and bystander ISF, and using a behavioral measure of costly punishment.

Here, I include a few minor suggestions that in my view would improve the paper:

- The term "bystander effect" is traditionally used in some fields of social sciences to describe a different phenomenon (namely a lower likelihood of intervening in groups). This may lead to potential confusion, I was especially confused the first time I read this term in the abstract and the introduction, not understanding how it could emerge in such a setting. The authors should consider using a different term, such as "bystander ISE," to clarify this specific context.
- Figures 2a and 2b were personally difficult to understand and would benefit from additional explanation, particularly in the footnotes.
- The sample size for Canada is smaller. I wonder whether this is due to higher attrition rates or other factors. Given that much of the paper is about comparing the effects in different countries, it would be helpful to compare attrition rates across the three countries to ensure this does not bias the results.
- An additional analysis in the appendix could explore whether different messages trigger more intense intergroup sensitivity effects (ISE) or bystander ISE, and if so, why. This could provide further insights into the mechanisms driving these effects.

Reviewer #3 (Remarks to the Author):

This work examines responses to intergroup and intragroup criticism across three cultural groups: Canada, China, and Japan. Researchers also took into account the target of the criticism: One's own group or another group. Finally, they examined the potential role of individual differences in individualism, collectivism, and honor endorsement.

I applaud the researchers for their efforts to test ISE in understudied countries and for including a behavioral measure in addition to perception of criticism. I have a few comments and questions regarding the theoretical and methodological aspects of the paper.

The terminology of Western vs. Non-Western is less widely used nowadays, as cultural psychologists have discovered several cultural constructs that describe and distinguish societies beyond this East-West distinction. A different, perhaps a more specific terminology would be better.

Moreover, in the introduction section, the definition of the three cultural orientations can be elaborated more. For example, on page 6, honor cultures are described as endorsing more violent responses to transgressions; however, things are usually more nuanced than that (for example, see Uskul et al., 2023). As the main focus of this paper is to test ISE across different cultures, a more in-depth explanation of cultural orientations is much needed.

The choice of Canada, China, and Japan as research locations is explained by the fact that these societies represent an individualistic, a collectivistic, and an honor culture, respectively. I'm curious about why the researchers chose Japan as an honor culture. Similar to China, Japan is also a collectivistic culture. Moreover, it is a face culture, in which there is strong emphasis on humility, harmony, and hierarchy (according to the honor, face, and dignity framework; Leung & Cohen, 2011). Why, for example, did the researchers not choose a Middle Eastern or Latin American honor culture instead? If their goal was to tease apart the role of collectivism and honor, (by having two collectivistic cultures, one being a little more honor oriented than the other), then that should have been made more explicit in the paper.

The term bystander effect was used a few times, which at first confused me a little because of the term that's used in prosocial behavior literature. Consistently using the term "bystander ISE" or using a different term such as observer effect could prevent confusion.

The researchers frame their work as an examination of criticism, however, when I read the comments they used in the study, I thought they sounded more like an insult to me (maybe I'm showing my honor culture background 😊). I understand that choosing harsher criticisms is a conservative way of testing their hypotheses but in my opinion, the researchers should revise their terminology and call it intergroup insult or hostility instead of criticism.

Participants always answered the group identification questions after reading and responding to the critical comments, some which are about their own group. Might this order have affected participants' identification level? For example, some participants' identification may have become stronger because of defensiveness, whereas others may have wanted to distance themselves from their group after reading about the negative comments. I think a 2-stage design would have been more ideal, in which participants would complete the group identification, as well as the cultural orientation measures at time 1, and perhaps a week or so later, the critical comments at time 2.

The researchers could also talk more about the implications of their findings. Finding cultural differences is interesting enough, but how can we use this information to reduce hostile reactions to criticism coming from the outgroup. What kind of values (that are perhaps already more prevalent in some cultures than in others) can be promoted to reduce these reactions?

I may have missed the following details but;

What was the gender of the message source? Was it matched with the participant gender? And in general, did the researchers take into account participant gender in their analyses? Why? Why not?

What was the racial and ethnic background of Canadian participants?

Were the critical comments pilot-tested?

I enjoyed reading your paper and found it interesting. I hope these comments help. Best of luck!

EDITORIAL POLICIES

We ask that you ensure your manuscript complies with our editorial policies and reporting requirements.

To that end, we require revised manuscripts to be accompanied by two completed items: a reporting summary that collects information on study design and procedure, and an editorial policy checklist that verifies compliance with all required editorial policies.

- <https://www.nature.com/documents/nr-reporting-summary.zip>>Nature Research Reporting Summary
- <https://www.nature.com/documents/nr-editorial-policy-checklist.pdf>>Editorial Policy Checklist

All points on the policy checklist must be addressed. Your revised manuscript can only be sent back to the referees if these checklists are completed and uploaded with the revision.

Notes: If you have submitted a Stage 1 Registered Report, Review, Primer, Comment, or Perspective you do not need to

submit these forms. If you have already submitted these forms, you may disregard this request.

If you experience problems in linking your ORCID, please contact the Platform Support Helpdesk.

Version 1:

Decision Letter:

Dear Lukas,

Your manuscript titled "Message Source Effects on Rejection and Costly Punishment of Criticism Across Cultures" has now been seen by our reviewers, whose comments appear below. In light of their advice I am delighted to say that we are happy, in principle, to publish a suitably revised version in Communications Psychology.

We therefore invite you to revise your paper one last time to address the remaining concerns of our reviewers and a list of editorial requests. At the same time we ask that you edit your manuscript to comply with our format requirements and to maximise the accessibility and therefore the impact of your work.

EDITORIAL REQUESTS:

Your protocol was comprehensively preregistered and peer-reviewed within the ZPID's framework. To comply with our preregistration policy (<https://www.nature.com/commspsychol/editorial-policies/preregistration-policy>; which is aligned with reporting instructions in your preregistration protocol), we require that hypotheses and methods are reported as preregistered. Your preregistration protocol includes a detailed methods description that should serve as a template for the paper's Methods section (which is currently too brief and needs to be moved to appear between the Introduction and Results). More detailed guidance is enclosed in the attached "Editorial Request Table".

Please review our specific editorial comments and requests regarding your manuscript in the attached Editorial Requests Table. Please outline your response to each request in the right hand column. Please upload the completed table with your manuscript files as a Related Manuscript file.

SUBMISSION INFORMATION:

OPEN ACCESS:

Link Redacted

Best wishes,

Marike

Marike Schiffer, PhD
Chief Editor
Communications Psychology

REVIEWERS' COMMENTS:

Reviewer #1 (Remarks to the Author):

The authors addressed my comments adequately and, in my view, the paper is ready for publication. I'd like to congratulate the authors for a nice, interesting, and well-written paper.

Reviewer #3 (Remarks to the Author):

Thank you for addressing my comments so carefully. I just have a couple of notes.

On pages 5 and 18, the authors describe Japan as an "honor (face) culture." The parentheses give the impression that the terms honor and face are interchangeable, but they are not the same, as summarized by Leung & Cohen (2011). Japanese culture entails components of both cultural logics (whereas China is primarily a face culture). I think the authors should explain this distinction to prevent confusion.

On page 6: "... honor entails identifying hierarchical relations and maintaining/improving one's social standing." This is a more suitable description of face cultures, especially the hierarchy part. Again, similar to my first point, a clear definition of honor and face logics is needed even if it's short.

Response to Reviewer 1 Comments

Reviewer 1:

The paper explores how people across different cultures react to criticism of their national group when the criticism comes from within their group (intragroup) versus from outside their group (intergroup). The study was conducted with participants from China, Canada, and Japan, representing collectivistic, individualistic, and honor cultures, respectively. The researchers found that people generally reject criticism of their group more strongly when it comes from an outsider. This rejection also extends to criticism of other national groups, particularly in Canada and Japan.

The study is pre-registered, methodologically sound, and the analysis is well-developed. As someone coming from a different field, my take is that this study makes a significant contribution by comparing reactions across three different cultures in a high-powered study, exploring both direct and bystander ISE, and using a behavioral measure of costly punishment.

Author Response: Thank you for serving as a reviewer, your positive evaluation and your helpful feedback. We address each of the points you raised below.

Reviewer 1 Comment 1:

The term "bystander effect" is traditionally used in some fields of social sciences to describe a different phenomenon (namely a lower likelihood of intervening in groups). This may lead to potential confusion, I was especially confused the first time I read this term in the abstract and the introduction, not understanding how it could emerge in such a setting. The authors should consider using a different term, such as "bystander ISE," to clarify this specific context.

Author Response 1: Thank you for pointing out this ambiguity. We now consistently refer to "bystander ISE" throughout the manuscript.

Reviewer 1 Comment 2:

Figures 2a and 2b were personally difficult to understand and would benefit from additional explanation, particularly in the footnotes.

Author Response 2: We now elaborate on Johnson-Neyman analyses as follows:

Quote: "Figure 2a. Johnson-Neyman-Plots for the moderation analysis of the message source effect (ingroup vs. outgroup) for criticism of participants' own national group (classic ISE) conditional on reported vertical individualism [VI] on the y-axis. Note: The message source effect (difference between within-group source and between-group source) is plotted on the y-axis and the moderator (VI) is plotted on the x-axis. The message source effect is significant at $p < .01$ in green-shaded areas and non-significant in red-shaded areas. Bold horizontal line indicates observed values of vertical individualism. Bonus: Number of bonus lottery tickets assigned." (p. 14)

Reviewer 1 Comment 3:

The sample size for Canada is smaller. I wonder whether this is due to higher attrition rates or other factors. Given that much of the paper is about comparing the effects in different countries, it would be helpful to compare attrition rates across the three countries to ensure this does not bias the results.

Author Response 3: Thank you for pointing to this potential confound. Despite the smaller sample, the attrition in Canada was the lowest both in terms of the number of participants and the proportion of the sample. We have added this information as follows:

Quote: “Incomplete entries were fewest in Canada $n = 355$ (35%), the smallest of the three samples, and comparable in China $n = 989$ (56%), Japan = 741 (50%).” (p. 21)

Reviewer 1 Comment 4:

An additional analysis in the appendix could explore whether different messages trigger more intense intergroup sensitivity effects (ISE) or bystander ISE, and if so, why. This could provide further insights into the mechanisms driving these effects.

Author Response 4: All comments were designed by incorporating generic negative words to maximize experimental control. Accordingly, potential differences between responses to these comments would be hard to interpret. What is more, including the Comment Content factor would necessitate analyzing the Source and Target factors at the between participants' level. This is why such an exploratory analysis would have exceptionally low power, especially after controlling for repeated tests. Accordingly, we refrain from conducting these analyses but discuss the possibility of investigating comment content in future studies systematically.

Quote: “We presented harsh critical comments that were identical across national populations to afford a high experimental control. Despite this highly conservative test, we observed consistent effects. One may argue that effects may even be more pronounced if comments were tailored to common stereotypes of that specific national group (e.g., portraying Americans as uncultured or Europeans as old-fashioned⁴¹). In fact, due to their generic character, the comments used in the present study may be perceived as serving the pragmatic function of derogating the targeted group (i.e., as insults^{73, 74}). In line with this view, the observed ISEs in this study had descriptively smaller effect sizes than those observed in past registered reports.^{19, 20} It should be noted, however, that consistent message source effects emerged nevertheless, pointing to the potential of interpreting the comments to be somewhat constructive. Exploring how differences in comment content affect responses to criticism clearly is a fruitful direction for future research.⁷⁵” (p. 18)

Response to Reviewer 2 Comments

Reviewer 2:

This work examines responses to intergroup and intragroup criticism across three cultural groups: Canada, China, and Japan. Researchers also took into account the target of the criticism: One's own group or another group. Finally, they examined the potential role of individual differences in individualism, collectivism, and honor endorsement.

I applaud the researchers for their efforts to test ISE in understudied countries and for including a behavioral measure in addition to perception of criticism. I have a few comments and questions regarding the theoretical and methodological aspects of the paper.

Author Response: Thank you for your thorough and constructive evaluation of our paper. Below, we respond to each of the points you raised.

Reviewer 2 Comment 1:

The terminology of Western vs. Non-Western is less widely used nowadays, as cultural psychologists have discovered several cultural constructs that describe and distinguish societies beyond this East-West distinction. A different, perhaps a more specific terminology would be better.

Author Response 1: Thank you for pointing out to this linguistic ambiguity. We now clarify the respective paragraph as follows:

Quote: “The key limitation to such a conclusion is that existing studies were almost exclusively conducted in Australia, the United States, and Europe. Only about 10% of the world population live in these areas and their populations may differ systematically from other populations.²⁸ Specifically, all three commonly studied populations are highly industrialized and score high on the cultural dimension of individualism. A few studies have investigated the rejection of criticism in samples outside the United States, Australia, and Europe,^{29, 30, 31, 32} but none of these studies systematically compared primarily individualistic cultural regions with regions representative of other cultural mindsets, observed costly behavioral responses, and/or considered potential cultural moderators. Humans ubiquitously cooperate,^{33, 34} and *culture* signifies how societies organize this cooperation, including when and how to punish transgressions.^{34, 35, 36, 37} Directly related to the employed behavioral measures of costly punishment, recent research indicates that individualistic cultures prioritize monetary over psychological incentives.³⁸ Testing the generalizability of costly rejection of intergroup criticism thus requires carefully replicating this candidate effect beyond samples from individualistic cultures.³⁹” (p. 4)

Reviewer 2 Comment 2:

Moreover, in the introduction section, the definition of the three cultural orientations can be elaborated more. For example, on page 6, honor cultures are described as endorsing more violent responses to transgressions; however, things are usually more nuanced than that (for example, see Uskul et al., 2023). As the main focus of this paper is to test ISE across different cultures, a more in-depth explanation of cultural orientations is much needed.

Author Response 2: Thank you for pointing this out. While being mindful to keeping the paper concise, we have extended these sections as follows:

Quote: “We provide such a test, following our peer-reviewed pre-registration protocol (see Method, for details). We selected three countries representative of the cultural dimensions of individualism (Canada), collectivism (China), and honor (Japan).^{36, 43} Despite their cultural differences, these countries are quite similar on other dimensions, such as their degree of industrialization. Moreover, Japan and China both typically score high on collectivism but Japan adheres to an honor (face) culture to a greater degree.⁴⁴ The selected countries thus provide a stringent test of the consequences of three cultural mindsets, individualism, collectivism, and honor, for responses to intergroup criticism.

The three cultural mindsets have been shown to moderate basic processes such as responses to reputational threats to one’s own group,⁴⁵ but also responses to general norm violations.⁴⁶ In short, collectivism entails identifying similarities and fitting in, individualism entails identifying differences and sticking out, and honor entails identifying hierarchical relations and maintaining/improving one’s social standing. According to a social identity perspective, collectivistic cultures could show a greater classic ISE (i.e., source effect regarding criticism targeting participant’s own group) as these cultures put a greater emphasis on incurring personal costs in the service of and for defending one’s group.^{34, 47, 48} According to a norm perspective, the overall ISE (i.e., source effect independent of the criticism target) could be greater in honor cultures that endorse more violent responses to transgressions,^{49, 50, 51, 52} especially when these transgressions are substantial (as is the case for intergroup criticism). Honor cultures seem to promote such retaliation because those engaging in it are socially rewarded⁵³ and insults towards one’s group are taken personally.^{54”} (p. 5)

Reviewer 2 Comment 3:

The choice of Canada, China, and Japan as research locations is explained by the fact that these societies represent an individualistic, a collectivistic, and an honor culture, respectively. I’m curious about why the researchers chose Japan as an honor culture. Similar to China, Japan is also a collectivistic culture. Moreover, it is a face culture, in which there is strong emphasis on humility, harmony, and hierarchy (according to the honor, face, and dignity framework; Leung & Cohen, 2011). Why, for example, did the researchers not choose a Middle Eastern or Latin American honor culture instead? If their goal was to tease apart the role of collectivism and honor, (by having two collectivistic cultures, one being a little more honor oriented than the other), then that should have been made more explicit in the paper.

Author Response 3: We indeed sought to rule out as many other factors as possible in this first cultural comparison. To this end, we selected two collectivistic cultures that differ in their endorsement of honor values (see quote above). This said, we believe that studies in MENA cultures would provide important new insights. We now discuss this direction for future research as follows:

Quote: “Moreover, Japan adheres to an honor (face) logic where positive self-views depend on one’s evaluation by others in established hierarchies that are largely cooperative. Finally, Japanese participants have been observed to be relatively open to criticism, even self-criticize, apparently as a means to self-improvement.⁷³ In contrast, Middle-

Eastern and North-African (MENA) cultures adhere to an honor logic where positive self-views depend on one's evaluation by others in competitive interactions. Studying the ISE in such honor contexts is a fruitful avenue for future research." (p. 18)

Reviewer 2 Comment 4:

The term bystander effect was used a few times, which at first confused me a little because of the term that's used in prosocial behavior literature. Consistently using the term "bystander ISE" or using a different term such as observer effect could prevent confusion.

Author Response 4: We appreciate your input regarding our wording. We now consistently refer to "bystander ISE" in the context of the rejection of intergroup criticism targeting a group that respondents do not belong to (e.g., on p. 2, Abstract; p. 4, Introduction)

Reviewer 2 Comment 5:

The researchers frame their work as an examination of criticism, however, when I read the comments they used in the study, I thought they sounded more like an insult to me (maybe I'm showing my honor culture background 😊). I understand that choosing harsher criticisms is a conservative way of testing their hypotheses but in my opinion, the researchers should revise their terminology and call it intergroup insult or hostility instead of criticism.

Author Response 5: You raise an interesting point here. In pursuing this line of work, we have often wondered where this line between insult and criticism lies. As soft-spoken, liberal individuals from very polite communities, we refrain from making any comments like the ones we study. At the same time, we observe that public discourse is increasingly taking this more critical (hostile?) form, leading us to believe that studying comments with this level of clarity represents an ecologically valid and theoretically sound contribution. In line with this view, acceptance of the critical comments was still higher in the ingroup source conditions, although overall means were lower than in previous research. This said, studying where the line between insults and criticism lies within different cultural contexts is an interesting avenue for future research. Accordingly, we sought to incorporate your advice in the following part of the discussion:

Quote: "We presented harsh critical comments that were identical across national populations to afford a high experimental control. Despite this highly conservative test, we observed consistent effects. One may argue that effects may even be more pronounced if comments were tailored to common stereotypes of that specific national group (e.g., portraying Americans as uncultured or Europeans as old-fashioned⁴¹). In fact, due to their generic character, the comments used in the present study may be perceived as serving the pragmatic function of derogating the targeted group (i.e., as *insults*^{73, 74}). In line with this view, the observed ISEs in this study had descriptively smaller effect sizes than those observed in past registered reports.^{19, 20} It should be noted, however, that consistent message source effects emerged nevertheless, pointing to the potential of interpreting the comments to be somewhat constructive. Exploring how differences in comment content affect responses to criticism clearly is a fruitful direction for future research.⁷⁵" (p. 18)

Reviewer 2 Comment 6:

Participants always answered the group identification questions after reading and responding to the critical comments, some which are about their own group. Might this order have affected participants' identification level? For example, some participants' identification may have become stronger because of defensiveness, whereas others may have wanted to distance themselves from their group after reading about the negative comments. I think a 2-stage design would have been more ideal, in which participants would complete the group identification, as well as the cultural orientation measures at time 1, and perhaps a week or so later, the critical comments at time 2.

Author Response 6: We now discuss this possibility as follows:

Quote: "Lastly, the exploratory moderators identification and cultural preferences were assessed after the dependent measures. It is thus possible that the experimental tasks influenced these measures. However, in our within-participants design, all participants viewed all comments from each comment source. Accordingly, it is unlikely that the experimental manipulations systematically influenced the measured moderators. To empirically rule out this possibility, future studies should employ 2-stage designs, measuring the moderators at T1 and assessing responses to experimental manipulations at T2." (p. 19)

Reviewer 2 Comment 7:

The researchers could also talk more about the implications of their findings. Finding cultural differences is interesting enough, but how can we use this information to reduce hostile reactions to criticism coming from the outgroup. What kind of values (that are perhaps already more prevalent in some cultures than in others) can be promoted to reduce these reactions?

Author Response 7: We have extended the implications section as follows:

Quote: "The current work contributes to understanding and predicting these mechanics so we can change what lies ahead. Specifically, we observed that participants from regions associated with collectivism "stayed in their lane," that is showed a message source effect in response to criticism targeting their own group (i.e., a large classic ISE) but not other groups (i.e., no bystander ISE emerged). Moreover, individuals reporting a high endorsement of vertical individualism, a cultural orientation towards hierarchical self-reliance, showed stronger rejection of outgroup criticism. This would suggest that promoting more communal and horizontal orientations should reduce the observed ISE. This speculation is in line with recent research showing that communal behaviors (apologizing for violating conversational norms and waiting to be invited to speak) reduced the ISE.^{24, 85} However, even under these conditions, significant source effects remained. Apparently, the rejection of intergroup criticism is a ubiquitous phenomenon that is hard to overcome." (p. 20)

Reviewer 2 Comment 8:

I may have missed the following details but;

What was the gender of the message source? Was it matched with the participant gender? And in general, did the researchers take into account participant gender in their analyses? Why? Why not?

What was the racial and ethnic background of Canadian participants?

Were the critical comments pilot-tested?

I enjoyed reading your paper and found it interesting. I hope these comments help. Best of luck!

Author Response 8: Regarding these additional points, we did not specify source gender (see quote below), did not observe consistent gender effects (second quote below), and did not assess participants' racial or ethnic background. The comments were directly adapted from a previous registered report to allow for a stringent cultural comparison. We now discuss this in the paper (see third and fourth quote below). Thank you again for serving as a reviewer on our submission, for your helpful input and your constructive feedback!

Quote: "The gender of the message source was not specified." (p. 8)

Quote: "Including participant gender as an exploratory factor had no consistent influence on the observed Target and Source effects. Only one small but significant Target × Participant Gender interaction emerged on message threat ratings, $p = 0.040$, $\eta^2 < 0.01$. The observed effects were thus largely unaffected by participant gender." (p. 13)

Quote: "All participants read four different critical comments that contained equally negative descriptions of a national group, directly adapted from a previous large-scale registered report.²⁰" (p. 7)

Quote: "The comments were directly adapted from a previous registered report²⁰ to allow for reliable intercultural comparisons." (p. 22)

Response to Reviewer 1 Comments

Reviewer 1 Comment:

The authors addressed my comments adequately and, in my view, the paper is ready for publication. I'd like to congratulate the authors for a nice, interesting, and well-written paper.

Author Response: Thank you for recommending our paper for publication and your service as a reviewer.

Response to Reviewer 3 Comments

Reviewer 3 Comment 1:

Thank you for addressing my comments so carefully. I just have a couple of notes.

On pages 5 and 18, the authors describe Japan as an “honor (face) culture.” The parentheses give the impression that the terms honor and face are interchangeable, but they are not the same, as summarized by Leung & Cohen (2011). Japanese culture entails components of both cultural logics (whereas China is primarily a face culture). I think the authors should explain this distinction to prevent confusion.

Author Response 1: Thank you for acknowledging our efforts and your additional feedback. We now cite Leung and Cohen and have changed these sentences as follows: “Japan and China both typically score high on collectivism but Japan adheres to an honor culture to a greater degree.^{44, 45}” (p. 5) and “Japan adheres to an honor (as well as face) logic where positive self-views depend on one’s evaluation by others in established hierarchies that are largely cooperative.⁴⁵” (p. 21)

Reviewer3 Comment 2:

On page 6: "... honor entails identifying hierarchical relations and maintaining/improving one's social standing." This is a more suiting description of face cultures, especially the hierarchy part. Again, similar to my first point, a clear definition of honor and face logics is needed even if it's short.

Author Response 2: We have clarified this point as follows: "In short, collectivism entails identifying similarities and fitting in, individualism entails identifying differences and sticking out, and honor entails maintaining/defending one's social standing in terms of self-worth and reputation.⁴⁸" (p. 6). Thank you again for serving as a reviewer.